# Dynamic control of sequential retrieval speed in networks with heterogeneous learning rules

**Maxwell Gillett[1], Nicolas Brunel[1,2]\***

[1]Department of Neurobiology, Duke University, Durham, United States; [2]Department of Physics, Duke University, Durham, United States

**Abstract** Temporal rescaling of sequential neural activity has been observed in multiple brain areas during behaviors involving time estimation and motor execution at variable speeds. Temporally asymmetric Hebbian rules have been used in network models to learn and retrieve sequential activity, with characteristics that are qualitatively consistent with experimental observations. However, in these models sequential activity is retrieved at a fixed speed. Here, we investigate the effects of a heterogeneity of plasticity rules on network dynamics. In a model in which neurons differ by the degree of temporal symmetry of their plasticity rule, we find that retrieval speed can be controlled by varying external inputs to the network. Neurons with temporally symmetric plasticity rules act as brakes and tend to slow down the dynamics, while neurons with temporally asymmetric rules act as accelerators of the dynamics. We also find that such networks can naturally generate separate 'preparatory' and 'execution' activity patterns with appropriate external inputs.

**\*For correspondence:**
nicolas.brunel@duke.edu

## eLife assessment

The authors provide a **valuable** analysis of what neural circuit mechanisms enable varying the speed of retrieval of sequences, which is needed in situations such as reproducing motor patterns. Their use of heterogeneous plasticity rules to allow external currents to control speed of sequence recall is a novel alternative to other mechanisms proposed in the literature. They perform a **convincing** characterization of relevant properties of recall via simulations and theory, though a better mapping to biologically plausible mechanisms is left for future work.

## Introduction

Timing is a critical component in the proper planning and execution of temporally extended motor behaviors. In behaviors consisting of a single motor action, it may be desirable to control the duration of its execution. In behaviors composed of multiple actions, the precise time interval between actions can be a key determinant in the success of the behavior. How can the duration of these intervals be flexibly controlled in a network of neurons?

A simple mechanistic hypothesis for the regulation of motor related timing intervals posits a specialized neural circuit with network dynamics that vary in speed as a consequence of differing levels of constant external input. Several network models utilizing external input as a means of speed control have been proposed to account for cortical and striatal dynamics observed during motor execution (*Murray and Escola, 2017*; *Wang et al., 2018*). To account for speed control in cortex, a recurrent neural network model has been trained to achieve temporal rescaling of network activity as a function of external input (*Wang et al., 2018*). However, this model relies on supervised learning rules that may not be biologically plausible, and cannot generalize to other speeds from training

on just one example timing interval. To explain speed control of sequential activity in striatum, a recurrent inhibitory network model has been proposed with a feedforward structure learned through anti-Hebbian plasticity (*Murray and Escola, 2017*). This model demonstrates transient winner-take-all dynamics, with short-term synaptic depression facilitating transitions in activity from one group of neurons to the next, and external input controlling the duration of each group's transient activation. While experimental evidence for the necessary type of depressive adaptation mechanism exists in the striatum, it may not be present in all cortical areas where rescaling of sequential activity is observed. Whether speed can be controlled in network models constructed using Hebbian learning without this mechanism remains unknown.

Network models with a connectivity generated by temporally asymmetric synaptic plasticity provide a potential framework for explaining how sequential activity can arise from local biologically plausible learning rules (*Sompolinsky and Kanter, 1986*; *Kleinfeld, 1986*). In both rate and spiking networks, the temporal statistics of sequential activity in networks using this type of rule qualitatively match experimental findings made over both short and long timescales of observation in multiple tasks with timing components (*Gillett et al., 2020*). However, the speed of sequential dynamics in these models is constrained by the choice of temporal offset in the learning rule and neuronal time constant, and cannot be modulated with external input.

The Hebbian rules explored in this work and previous studies are approximations of various forms of spike-timing dependent plasticity (STDP). The effects of STDP can be quantified through kernels that measure the change in excitatory postsynaptic potential size at a synapse (as a proxy for synaptic strength), as a function of the timing difference between pre and postsynaptic spikes. Experimentally, a large diversity of STDP kernels have been characterized across cortical, subcortical, and cerebellar structures (*Abbott and Nelson, 2000*; *Suvrathan et al., 2016*). Kernels measured in cortex and hippocampus typically, but not always, exhibit a temporal asymmetry, in which presynaptic activity must precede postsynaptic activity to elicit a positive change in synaptic strength (*Bi and Poo, 1998*; *Egger et al., 1999*). Theoretical studies have shown that this temporal asymmetry can be used to store and retrieve sequences of activity (*Jun and Jin, 2007*; *Liu and Buonomano, 2009*; *Fiete et al., 2010*; *Waddington et al., 2012*; *Zheng and Triesch, 2014*; *Okubo et al., 2015*; *Ravid Tannenbaum and Burak, 2016*; *Murray and Escola, 2017*; *Weissenberger et al., 2017*; *Theodoni et al., 2018*; *Pereira and Brunel, 2019*; *Tupikov and Jin, 2020*; *Gillett et al., 2020*). However, symmetric kernels, in which coincident activity leads to strengthening regardless of the order of pre and post-synaptic spikes, have also been observed in multiple contexts - with high frequency plasticity induction protocols in cortex (*Sjöström et al., 2001*), in hippocampal cultures in the presence of dopamine (*Zhang et al., 2009*), and at excitatory-to-excitatory synapses in hippocampal CA3 (*Mishra et al., 2016*). Hebbian learning rules that are temporally symmetric lead instead to the creation of fixed point attractors (*Hopfield, 1982*; *Amit and Brunel, 1997*; *Wang, 2001*; *Brunel, 2005*; *Pereira and Brunel, 2018*). It is not known to what degree temporal asymmetry varies across synapses at the scale of local networks, but analysis of a calcium-based plasticity model demonstrates that the degree of asymmetry can be controlled via adjustment of biophysical parameters (*Graupner and Brunel, 2012*). We hypothesize that variability in the temporal offset expressed at a synapse may be a key ingredient in permitting the control of retrieval speed, suggesting a potential new role for the observed heterogeneity in STDP kernels.

In this work, we explore a generalization of previously investigated temporally asymmetric learning to multiple temporal offsets that captures this heterogeneity. Specifically, we find that varying the temporal asymmetry of the learning rule across synapses gives rise to network mechanisms that allow for the control of speed as a function of external inputs to the network. We start by considering a network with a bimodal distribution of heterogeneity in the learning rule, resulting in two distinct populations: one with a symmetric learning rule, and one with an asymmetric rule. We characterize the effect of input strength on retrieval speed and quality in these networks with connectivity generated using linear and nonlinear synaptic plasticity rules. We also find that transitions between fixed-point attractor-like 'preparatory' periods and sequential 'execution' phases can be realized in this model by rescaling the magnitude of external input. Finally, we demonstrate that networks with a uniform distribution of heterogeneity lead to qualitatively similar findings.

## Results

### Degree of symmetry in learning rule determines retrieval speed

We explore a network model in which the firing rate dynamics of each neuron $r_i$ in a population of size $N$ is described by the equation

$$\tau \frac{dr_i}{dt} = -r_i + \phi \left( \sum_{j=1}^{N} J_{ij} r_j + I_i^{ext}(t) \right) \tag{1}$$

where $\tau$ is the time constant of firing rate dynamics, $J_{ij}$ is the connectivity matrix, $\phi(x)$ is a sigmoidal neuronal transfer function (see Methods), and $I_i^{ext}(t)$ describes the external input provided to each neuron at time $t$.

We follow a similar learning procedure as in *Gillett et al., 2020*. A sequence of $P$ random i.i.d standard Gaussian patterns $\xi_i^\mu$ is presented to the network and stored in network connectivity. This sequence of patterns modifies the strength of synaptic connections $J_{ij}$ from neuron $j$ to $i$ according to a Hebbian learning rule that transforms pre and post synaptic inputs into synaptic weight changes. The resulting connectivity matrix $J_{ij}$ is a generalization of previously studied rules which combines both temporally symmetric and asymmetric learning (*Pereira and Brunel, 2018*; *Gillett et al., 2020*),

$$J_{ij} = A \frac{c_{ij}}{Nc} \left( \sum_{\mu=1}^{P} z_i f(\xi_i^\mu) g(\xi_j^\mu) + \sum_{\mu=1}^{P-1} (1 - z_i) f(\xi_i^{\mu+1}) g(\xi_j^\mu) \right) \tag{2}$$

where $c_{ij}$ is a matrix describing the structural connectivity, whose entries are given by i.i.d. Bernoulli random variables, $p(c_{ij} = 1) = c$, $p(c_{ij} = 0) = 1 - c$, where $c$ is the connection probability; The functions $g(x)$ and $f(x)$ describe how the synaptic plasticity rule depends on pre and postsynaptic input patterns during learning, respectively; The parameter $A$ controls the overall strength of the recurrent connections; And $z_i \in [0, 1]$ describes the degree of temporal symmetry at synapses of neuron $i$. A neuron with fully temporally symmetric plasticity is described by $z_i = 1$, while $z_i = 0$ indicates a neuron with fully temporally asymmetric plasticity. Note that we focus here to the case of a single sequence stored in synaptic connectivity, but such networks can also store multiple sequences (*Gillett et al., 2020*).

We first explore the bilinear learning rule scenario ($f(x) = g(x) = x$) with homogeneous synaptic plasticity, i.e. $z_i = z$ for all $i = 1, \ldots, N$. At the two extremes of this variable we can recover previously studied learning rules. When $z = 0$, only the second term in *Equation 2* is present, resulting in a purely temporally asymmetric rule. Networks with connectivity constructed using such a rule can recall a sequence of stored patterns, and their sequential retrieval dynamics have been extensively characterized (*Gillett et al., 2020*). When $z = 1$, synaptic plasticity is temporally symmetric, potentially leading to fixed point attractor dynamics (*Pereira and Brunel, 2018*). If $z$ is instead fixed to a value between 0 and 1, then the asymmetric component in the plasticity rule leads to the retrieval of the whole sequence, but the speed at which the sequence is retrieved strongly depends on $z$. For instance, in *Figure 1b* we demonstrate retrieval for an intermediate value of $z = 0.5$. Retrieval is quantified by plotting the Pearson correlation of the instantaneous firing rate $r(t)$ with each stored pattern $\xi^\mu$ as a function of time (see Methods). During sequence retrieval, correlations with individual patterns in the sequence increase, peak and decrease one after the other, indicating the network transiently visit states close to each of the patterns in succession. We find that in such a network, retrieval speed strongly depends on $z$. For the parameters in *Figure 1b*, retrieval proceeds nearly twice as slowly as compared to a network with connectivity arising from a purely asymmetric learning rule, where retrieval speed is fixed by the time constant of the firing rate dynamics (*Gillett et al., 2020*). However, retrieval speed is fixed by the choice of $z$ (see *Figure 1c* showing a linear dependence of speed on $z$), and cannot be dynamically modulated in response to changes in the external input $I_i^{ext}$.

### Heterogeneity in synaptic plasticity temporal asymmetry gives rise to a speed control mechanism

We next explored whether adding heterogeneity to this learning rule, allowing $z_i$ to differ across synapses, can produce networks capable of both recalling stored sequences of patterns and modulating the speed of recall. We initially consider a bimodal distribution of degrees of temporal symmetry across the network. For each neuron, $z_i$ was drawn randomly and independently as a Bernoulli random

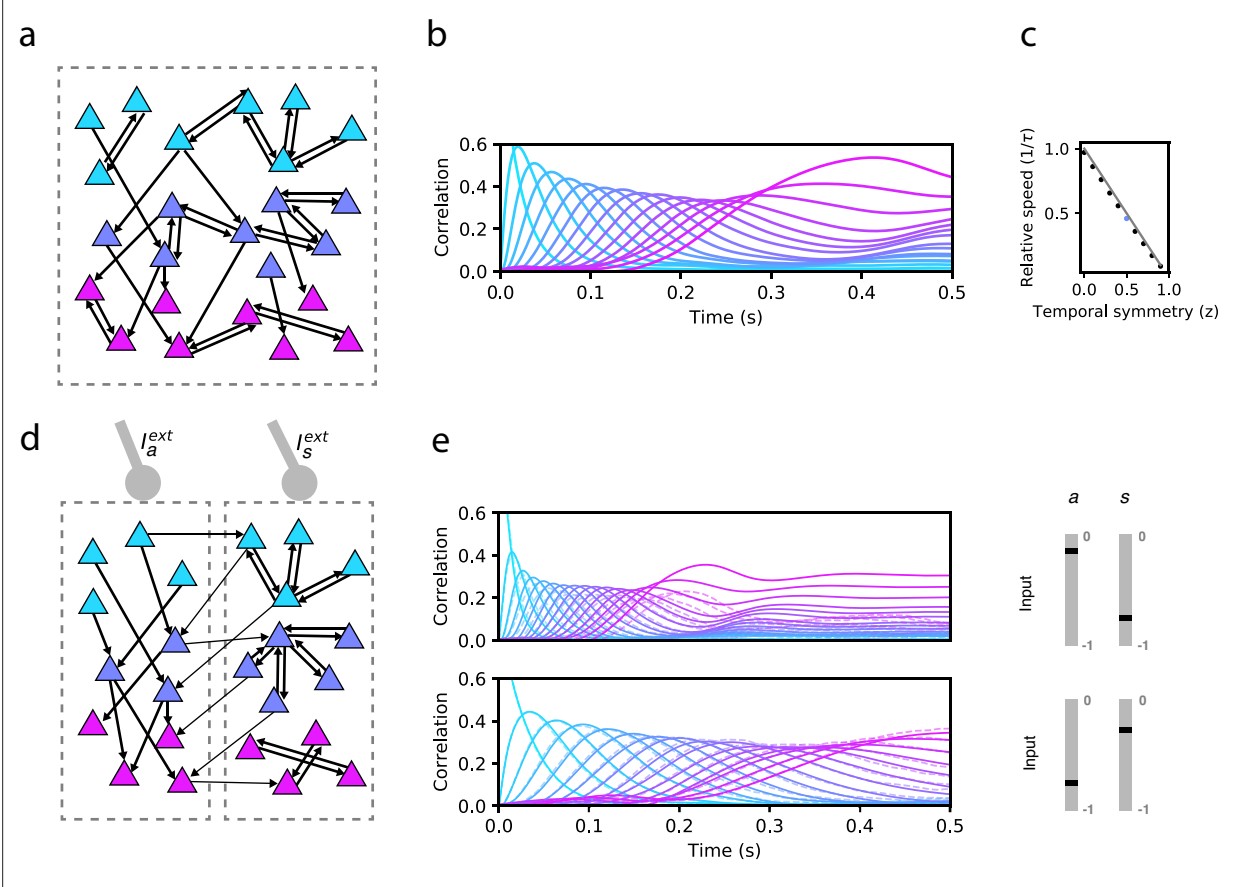

**Figure 1.** Network model and sequence retrieval. (**a**) Schematic of network connectivity after learning with a plasticity rule that combines temporally symmetric and asymmetric components. The network stores a sequence of patterns that activate non-overlapping sets of neurons (colored according to the pattern that activates them). Note connections both within each set, and from one set to the next. (**b**) Correlation of each stored pattern with network activity following initialization to the first pattern. Retrieval speed is fixed by the balance of symmetry/asymmetry at the synapse. (**c**) Relative retrieval speed as a function of temporal symmetry (**z**), showing linear relationship. Solid line: $1 - z$, the speed computed from MFT (see Methods). Black dots: Network simulations. (**d**) Connectivity of a network with two types of neurons, asymmetric (left) and symmetric (right). Note that the connections from left neurons project to neurons active in the next pattern in the sequence, while connections from right neurons project to neurons active in the same pattern as the pre-synaptic neuron. The two types of neurons can be driven differentially by external inputs ($I_a^{ext}$ and $I_s^{ext}$, respectively) (**e**) Solid lines: correlations as in (**a**) for two distinct pairs of input strengths (in the range [–1,0] for $I_a^{ext}$ and $I_s^{ext}$), demonstrating two different retrieval speeds. Dashed lines: correlations with noisy time-dependent heterogeneous input added to the network (see Methods). In the simulations shown on the center and right panels, $N = 80,000$, $c = 0.005$, $\tau = 10$ms, $P = 16$, $A = 2$, $\theta = 0$, and $\sigma = 0.1$. For simplicity, we depict only 3 of the 16 stored patterns in the left schematics.

variable with probability $p(z_i = 1) = 0.5$, $p(z_i = 0) = 0.5$. As a result, the network of $N$ neurons can be divided into two subpopulations of approximately equal sizes $N_a = N_s = \frac{N}{2}$ neurons, according to the learning rule present at their synapses:

$$\tau \frac{dr_i^a}{dt} = -r_i^a + \phi \left( \sum_{j=1}^{N_a} J_{ij}^{aa} r_j^a + \sum_{j=1}^{N_s} J_{ij}^{as} r_j^s + I_a^{ext}(t) \right) \tag{3}$$

$$\tau \frac{dr_i^s}{dt} = -r_i^s + \phi \left( \sum_{j=1}^{N_s} J_{ij}^{ss} r_j^s + \sum_{j=1}^{N_a} J_{ij}^{sa} r_j^a + I_s^{ext}(t) \right) \tag{4}$$

where the connectivity matrix is given by

$$J_{ij}^{aX} = \frac{c_{ij}^{aX}}{N_a c} \sum_{\mu}^{P-1} f(\xi_i^{a,\mu+1}) g(\xi_j^{X,\mu}) \tag{5}$$

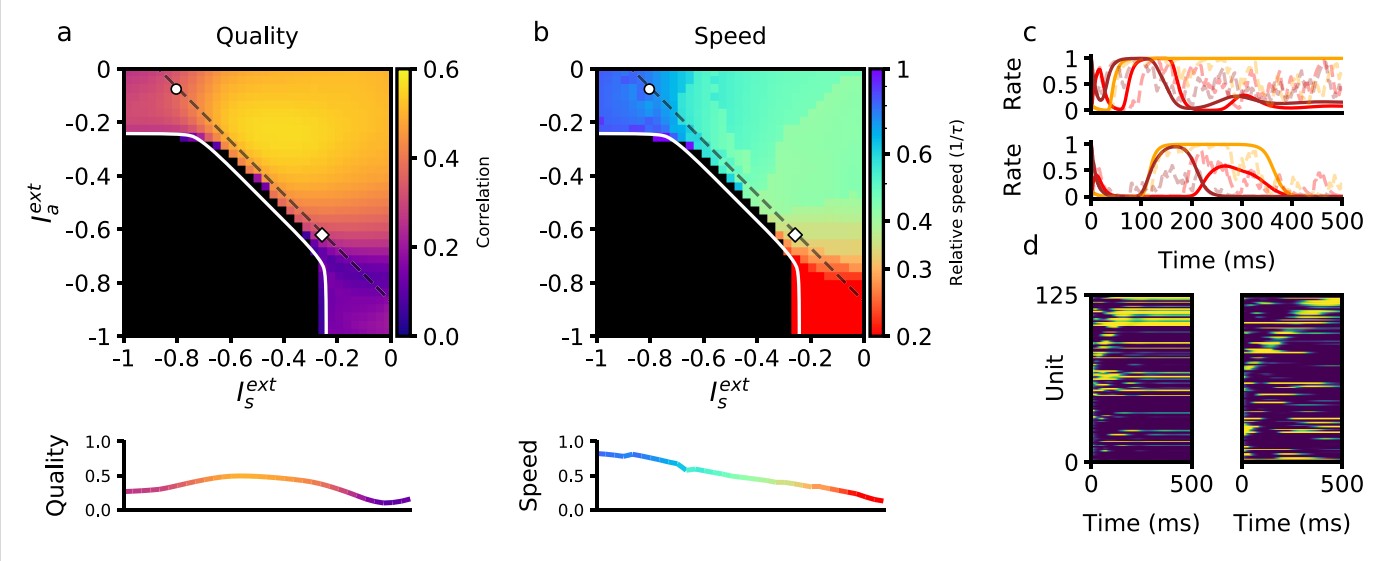

**Figure 2.** Retrieval properties depend on external inputs. (**a**) Retrieval quality, defined as the peak correlation $m^P$ of the final pattern in the sequence, as a function of external inputs to asymmetric population $I_a^{ext}$, and to symmetric population $I_s^{ext}$. The white line bounds the region of successful retrieval. Below this line (black region), retrieval is not possible, regardless of initial condition (see Methods). (**b**) Retrieval speed, measured by averaging the inverse time between consecutive pattern correlation peaks (see Methods). (**c**) Solid lines: firing rates of three randomly selected neurons during retrieval for parameters corresponding to the circle (left) and diamond (right) in panels (**a–b**), which are the same parameters used in *Figure 1e*. Note the approximate (but not exact) temporal scaling by a factor ~ 3 between these two sets of external inputs. Dashed lines: firing rates in response to the same noisy inputs as in *Figure 1e*. (**d**) Activity of 125 units (from a total of 80,000), sorted by peak firing rate time, for parameters corresponding to the circle (left) and diamond (right) in panels (**a–b**). All other parameters are as in *Figure 1b*.

$$J_{ij}^{sX} = \frac{c_{ij}^{sX}}{N_s c} \sum_{\mu}^{P} f(\xi_i^{s,\mu}) g(\xi_j^{X,\mu}) \tag{6}$$

and where $X = a, s$ denotes the presynaptic population. Note that the external input $I_X^{ext}$ now depends on the population. To reduce the space of possible learning rules, we have assumed that the type of learning at a synapse depends only on the identity of the postsynaptic neuron. The bimodal distribution of $z_i$ restricts synapses to only one of the two types of plasticity, but in the final section entitled 'Retrieval with a broad distribution of learning rules' we relax this constraint.

In *Figure 1e*, we show an example of how the stored sequence can be retrieved under different input conditions. In both the top and bottom panels of 1e, network activity is initialized to the first pattern in the sequence, and a constant external input is provided to each subpopulation ('asymmetric' input $I_a^{ext}$, and 'symmetric' input $I_s^{ext}$). In the top panel, the symmetric population is effectively silenced with strongly negative input, resulting in retrieval that lasts approximately $\tau P$, consistent with the dynamics being driven purely by the asymmetric component in the learning rule (*Gillett et al., 2020*). In the bottom panel, this input is no longer strongly negative, causing retrieval time to more than double, due to the effect of the symmetric population that tends to slow down the dynamics. Retrieval in both conditions is robust to noise, as shown in *Figure 1E*, in which noisy inputs to neurons strongly perturb single neuron firing rates but leave sequence retrieval intact at both speeds (see Methods).

To characterize how retrieval time depends on these two sources of external input, we explored the space of the parameters defining the inputs to the network, $I_a$ and $I_s$. In *Figure 2*, we show the dependence of retrieval quality and speed on these variables. Retrieval quality is quantified by measuring the maximal correlation of the final pattern in the retrieved sequence. Retrieval speed is measured in units of the inverse of the neural time constant, $\tau^{-1}$. It is computed by measuring the average inverse time between the peaks of consecutive correlations of the network state with consecutive patterns in the sequence. For example, a speed of 0.5 corresponds to an average time difference of $2\tau$ between the peaks of the correlations of two consecutive retrieved patterns with network state. In the upper left quadrant of *Figure 2b*, speed depends primarily on the strength of input to the

symmetric population. Moving away from this region in the direction of increasing symmetric input, retrieval speed slows down to approximately 0.5. In the lower right quadrant, retrieval speed instead depends primarily on the strength of external input provided to the asymmetric population. As this negative input grows, retrieval speed becomes approximately four times slower than the speed of the purely asymmetric network. In *Figure 2*, we have focused on the region in which external inputs are negative. This is because in our model external inputs are expressed relative to the threshold, and this region leads to biologically plausible average firing rates that are much smaller than the maximal firing rates (see Methods). While we have focused on negative input in *Figure 2*, retrieval speed is also modulated by positive input. Interestingly, it is the magnitude, not sign, of the input that determines retrieval speed. Expanding the phase diagram in panel (b) to positive input shows that the same dependence holds: values for retrieval speed are approximately symmetric about the $I_a^{ext}$ and $I_s^{ext}$ axes (not shown).

## Flexible retrieval with a non-linear plasticity rule

We next considered the consequences of a nonlinear learning rule implemented by the following presynaptic and postsynaptic functions in *Equation 2*:

$$f(x) = q_f - 1 + \Theta(x - x_f) \tag{7}$$

$$g(x) = q_g - 1 + \Theta(x - x_g) \tag{8}$$

where $\Theta(x)$ is the Heaviside function. This rule binarizes the activity patterns $\xi$ according to a threshold, and its effects on persistent and sequential network activity have been studied extensively (*Lim et al., 2015*; *Pereira and Brunel, 2018*; *Gillett et al., 2020*). The parameter $q_g$ is chosen such that $\int Dzg(z) = 0$, which keeps the mean connection strength at zero. The general dependency of retrieval speed on asymmetric and symmetric inputs in a network utilizing this rule is similar to that of the bilinear rule (see *Figure 2*). One key difference is that a much wider range of speeds can be achieved using a nonlinear rule within the same retrieval quality bounds (see Methods). In fact, retrieval speed can now be arbitrarily slowed down, and even completely stopped when the input to the asymmetric population is sufficiently negative (see white dots in *Figure 3b*). In this region, persistent activity is stable, and there exists a fixed point attractor correlated with any of the patterns in any stored sequence. There also exists a region in which sequential activity stops in the middle of

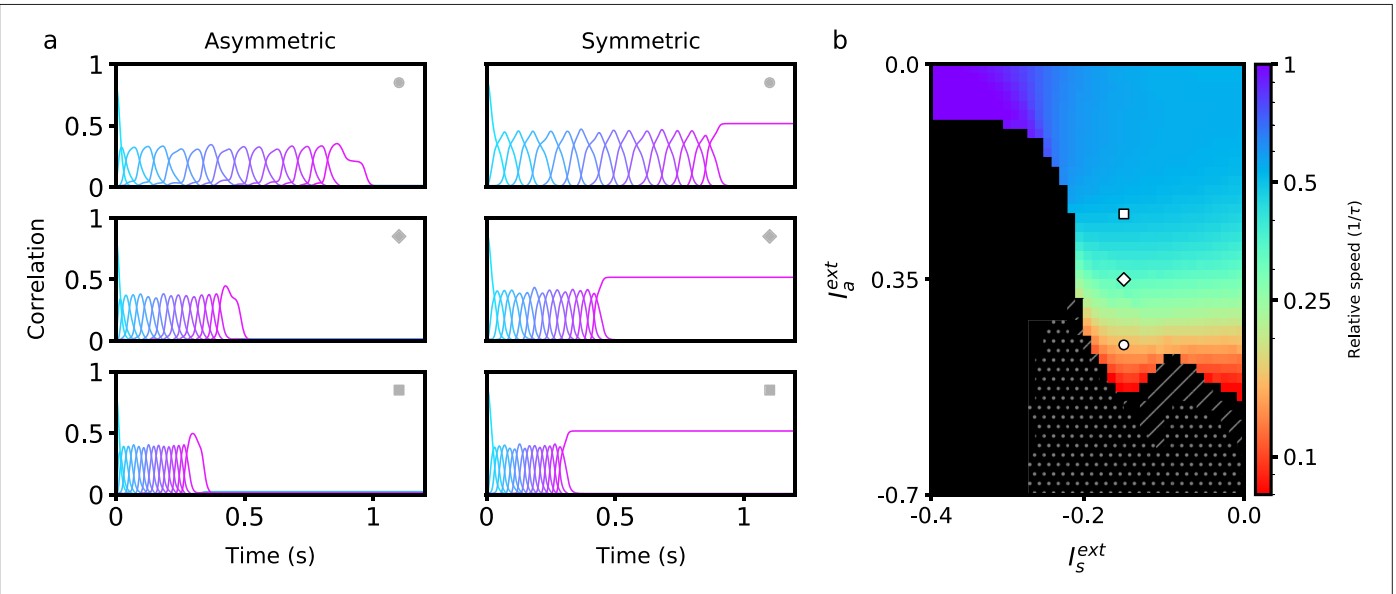

**Figure 3.** Retrieval properties in networks with nonlinear learning rules. (**a**) Correlations between stored patterns and network states in asymmetric (left) and symmetric (right) populations, for three different external input combinations (denoted by the inset symbol, see right panel). (**b**) Retrieval speed as a function of parameters describing external inputs, similarly as in *Figure 2*. White dots indicate the region in which stable persistent activity of the first pattern is present. Hatched diagonal lines indicate the region in which incomplete sequential activity terminates in stable persistent activity. All parameters are as in *Figure 2*, except $A = 20$ and $\sigma = 0.05$. The parameters of the learning rule are $x_f = 1.5$, $x_g = 1.5$, $q_f = 0.8$, and $q_f = 0.8$.

retrieval and switches to stable persistent activity (see hatched diagonal lines in *Figure 3b*). Note that retrieval is not considered to be successful in this region (as the sequence is not fully retrieved), and so it is plotted in black.

### Temporally varying external inputs can lead to transitions between persistent and sequential activity

We next explored how this heterogeneity might be used not only to control the speed of dynamics, but also to trigger transitions between qualitatively different dynamics. In *Figure 4*, we use the same nonlinear model as in the previous section, and present discrete, time-dependent inputs intended to achieve persistent retrieval of a single pattern, followed by sequential retrieval of the remaining patterns at a specified time. To initiate persistent activity, we briefly present the first pattern as an input to the symmetric population. This elicits persistent activity in this population, as reflected by the sustained positive correlation of the symmetric population with the first pattern during the first 200ms (*Figure 4b*). This activity does not recruit sequential activity in either population, however, as the asymmetric population responsible for that transition is presented with sufficiently strong negative input during this period. To initiate sequential activity, inhibition to the asymmetric population is released after $t = 0.2$ s, prompting the network to retrieve the stored sequence in both populations.

Note that in this scenario also, a sequence can be retrieved at various speeds, using the same inputs during the persistent period, but changing the level of constant stimulation provided during retrieval (compare left and right panels in *Figure 4b*). As in a network with only a single asymmetric population, single neuron activity in this network is temporally sparse, with many neurons being active only at specific time intervals (*Figure 4c*).

In our network, stability of persistent activity requires the dependence of the plasticity rule on pre and/or post synaptic firing rates to be non-linear. With a bilinear learning rule and Gaussian patterns, the network dynamics does not converge to fixed-point attractors that are correlated with a single pattern, but rather to mixed states correlated with multiple patterns (*Amit et al., 1985*).

The dynamics shown in *Figure 4* reproduces some of the landmark features observed in electrophysiological recordings during delayed motor tasks. In such tasks, a preparatory period follows presentation of a cue (e.g. instructing a target direction or a desired response speed), during which the animal can prepare the motor response, but not execute it (*Churchland et al., 2012*). This period is typically characterized by persistent activity of specific groups of neurons, whereas during motor execution those same neurons instead display transient activity (*Svoboda and Li, 2018*).

### Flexible sequence retrieval in networks with a continuous distribution of degrees of temporal symmetry

Up to this point, we have analyzed a network model in which neurons are separated in two discrete classes distinguished by their plasticity rule (symmetric or asymmetric). For a given postsynaptic neuron, the learning rule present at all presynaptic synapses was chosen to be either temporally symmetric or asymmetric with equal probability, defining two distinct subpopulations of neurons. Can retrieval speed still be modulated by external input when synapses do not fall into such a binary classification, but have more heterogeneous properties? To model this heterogeneity, we chose to embed a continuum of learning rules. Instead of a bimodal distribution for $z_i$ in *Equation 2*, we choose a uniform distribution on the interval $[0, 1]$. The input $I_i^{ext}$ provided to each neuron $i$ in *Equation 1* is a linear combination of symmetric and asymmetric input components: $I_i^{ext} = z_i I_s^{ext} + (1 - z_i) I_a^{ext}$. We also choose to investigate a network with the previously described non-linear plasticity rule. *Figure 5* shows that a network with these modifications also exhibits flexible sequence retrieval, and that speed decreases as the asymmetric input component becomes more negative. However, as shown in *Figure 5c*, to reach slower speeds a positive $I_s^{ext}$ is now required. Note that a region of stable persistent activity is no longer present in this scenario, as stable persistent activity requires that a finite fraction of neurons in the network have a symmetric plasticity rule.

### Learning external input strengths using a reward-based plasticity rule

The low-dimensional external inputs used to regulate speed are unrelated to the stored sequential input patterns. This suggests that a mapping from external inputs to retrieval speed can be learned independently from a particular set of sequential patterns. We demonstrated that a reinforcement

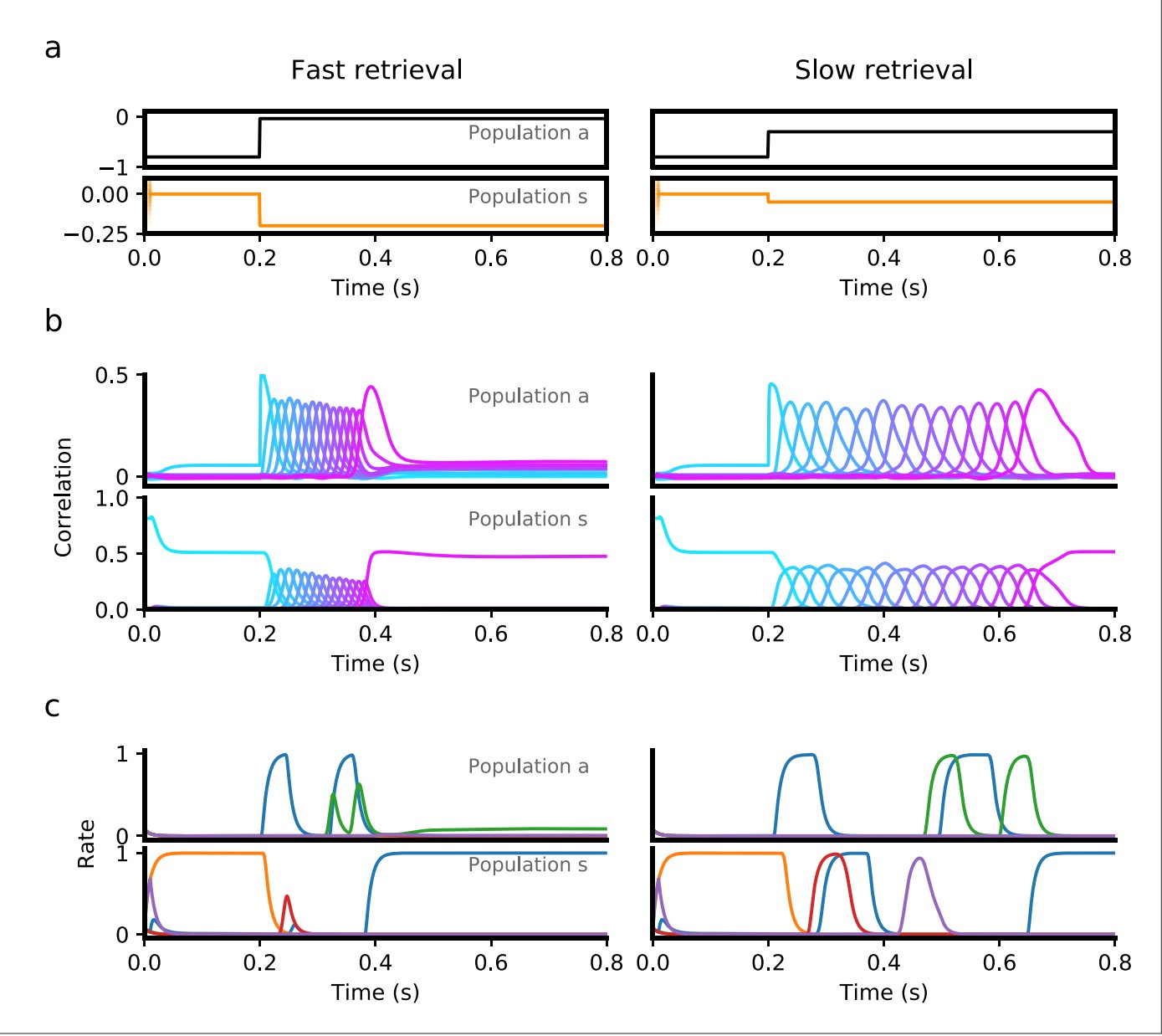

**Figure 4.** Transition from persistent activity ('preparatory' period) to sequence retrieval ('execution' period) mediated by external input. (**a**) Inputs provided to the asymmetric (black) and symmetric population (orange) consist of a 'preparatory period' input lasting 200ms, followed by an 'execution period' input that is fixed for the rest of the interval. During a 200ms preparatory period, a brief input is presented to the symmetric population for the first 10ms, which drives the network to a state which is strongly correlated with the first pattern in a sequence. This input is removed after 10ms, but the network remains in a persistent activity state corresponding the the first pattern, because a strong negative input is presented to the asymmetric population throughout the entire 200ms, which prevents the network from retrieving the sequence. At the end of this period, the input to the symmetric population is decreased, while the asymmetric population is increased, which leads to retrieval of the sequence ('execution period'). Sequence retrieval can happen at different speeds, depending on the inputs to the asymmetric and symmetric populations. (**b**) Correlations with stored patterns in the sequence in each population, in each input scenario. Note correlations in the slow retrieval case are temporally scaled by a factor ~ 2.5 compared to the fast retrieval case. (**c**) Example single unit firing rates in each population. Note that for some neurons firing rates do not follow a simple temporal rescaling - for instance the purple neuron in the symmetric population is active at around $t = 0.45$ in the slow retrieval case, but is not active in the fast retrieval case. All parameters are as in *Figure 3*, except $\theta = 0.07$ and $\sigma = 0.05$.

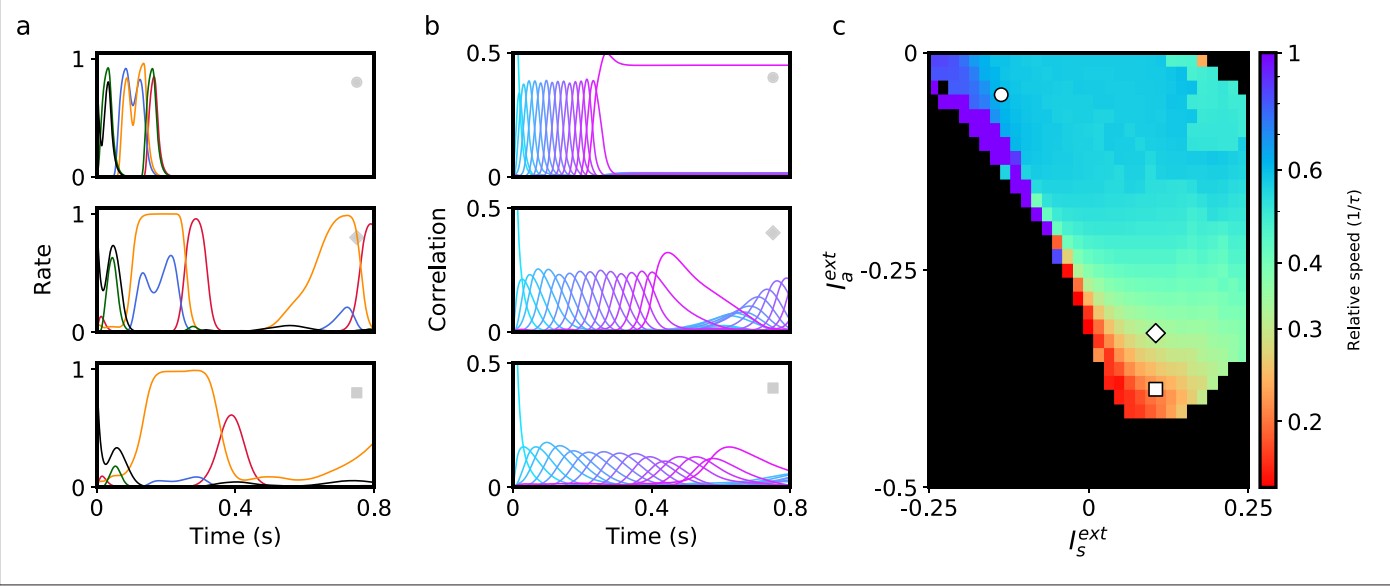

**Figure 5.** Retrieval of sequences in networks with heterogeneous learning rules described by a continuum of degrees of symmetry. (**a**) Firing rate dynamics of five representative neurons during retrieval for each external input configuration (see inset symbols in panel c). (**b**) Correlation of network activity with each stored pattern during retrieval for each external input configuration. (**c**) Retrieval speed as described in *Figure 2*. All parameters are as in *Figure 3* except $A = 10$.

learning rule can be used to converge to external input values implementing a desired speed (*Figure 6*). By using a reward signal measuring how similar retrieval is to the desired speed, the rule adjusts initially random external inputs to the appropriate values over the course of multiple trial repetitions (see Methods for details). Critically, once these external input values are learned, they can be used to modulate the retrieval speed of other stored sequences without having to relearn this mapping.

## Flexible retrieval of sequences in a spiking network

We have until now focused exclusively on rate networks that do not obey Dale's law. We now turn to networks composed of excitatory and inhibitory spiking neurons, as a more realistic model of neurobiological networks. We implemented learning in excitatory to excitatory synaptic connectivity, generalizing the procedure described in *Gillett et al., 2020* to two excitatory subpopulations. We found that successful speed control can be obtained in such networks using biases in external inputs to symmetric and asymmetric populations, as in the simpler rate model described above. *Figure 7* shows network simulations using two different external input configurations, leading to sequence retrieval at two different speeds. Interestingly, small external input biases ($\sim 1\text{mV}$) relative to the difference in spiking threshold and resting potential ($20\text{mV}$) are sufficient to generate a temporal rescaling of as large as $\sim 2$.

## Discussion

In this paper, we have introduced a new mechanism for flexible control of retrieval speed in networks storing sequences. This mechanism relies on heterogeneity of synaptic plasticity rules across neurons in the network, with different degrees of temporal asymmetry. Neurons with temporally symmetric plasticity act as brakes of the dynamics, as they stabilize network activity in its current state, while neurons with temporally asymmetric plasticity act instead as accelerators, as they push the network toward the next pattern in the sequence. The speed of retrieval can then be modified in a flexible way by changing external inputs driving these two types of neurons. Furthermore, we found that this mechanism can be used to gate transitions between persistent and sequential activity. We showed that appropriate inputs can be learned using a reinforcement learning scheme. Finally, we also showed

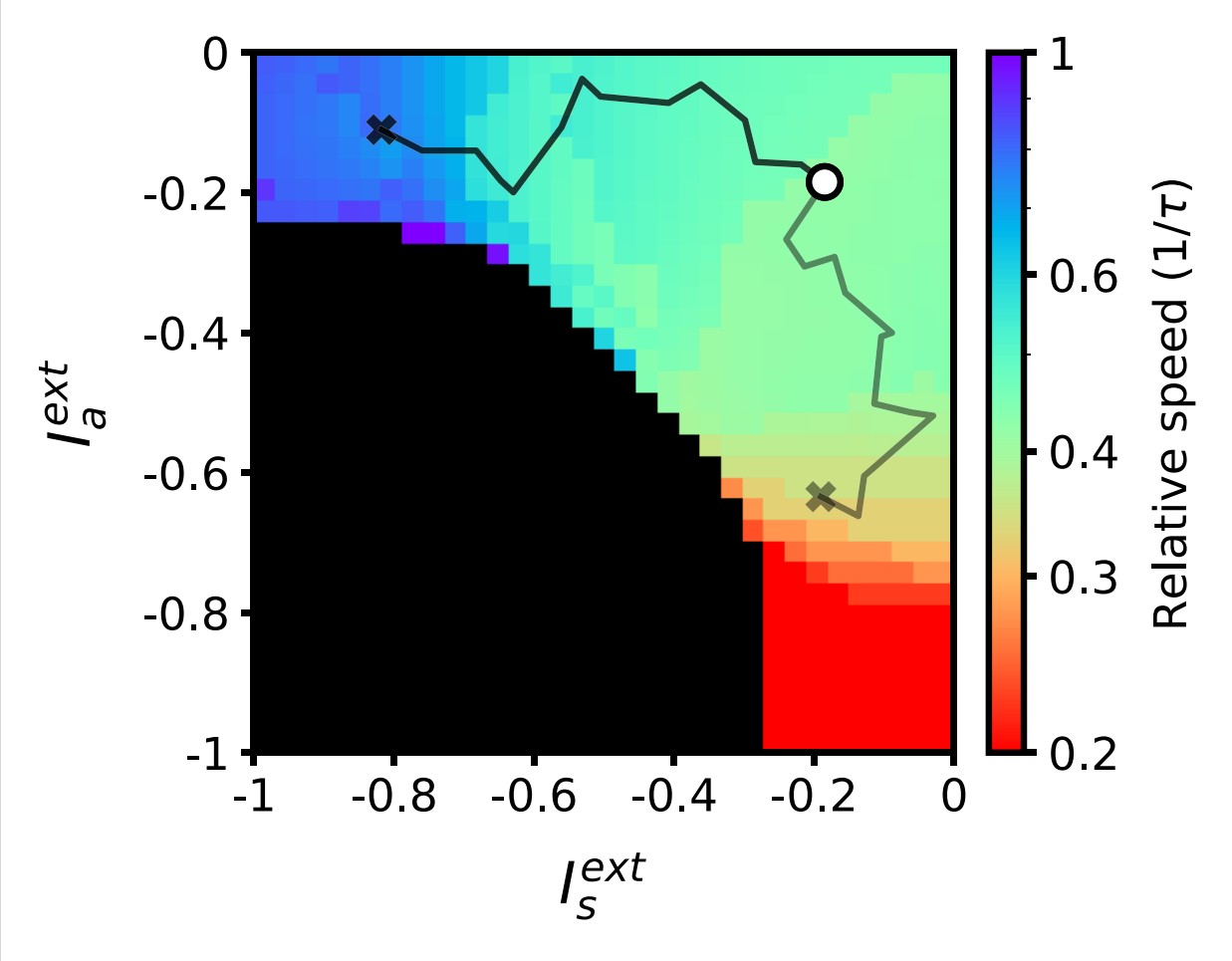

**Figure 6.** Desired target speeds can be reached by adjusting external inputs using a reward-based learning rule. The black and grey lines denote the trajectories for two learning trials targeting different speeds. External inputs start at −0.2 (marked with a circle) and terminate at values implementing desired target speeds of 0.8 and 0.3 (marked with crosses). All parameters are as in **Figure 2**.

that networks of spiking neurons can generate the same behavior, provided the excitatory network is subdivided in asymmetric and symmetric neurons.

## Heterogeneity of synaptic plasticity

Our findings suggest a potential functional role for the experimentally observed diversity in synaptic plasticity rules (**Bi and Poo, 1998**; **Abbott and Nelson, 2000**; **Sjöström et al., 2001**; **Mishra et al., 2016**; **Suvrathan et al., 2016**). In particular, a wide diversity of spike-timing dependent plasticity (STDP) curves have been reported in various brain structures, and sometimes in the same structure. In the hippocampus, temporally asymmetric STDP is typically observed in cultures (**Bi and Poo, 1998**) or in CA3 to CA1 connections in slices in some conditions, but temporally symmetric STDP is observed in area CA3 (**Mishra et al., 2016**). Interestingly, the degree of temporal symmetry at CA3 to CA1 connections can be modulated by extracellular calcium concentration (**Inglebert et al., 2020**) and post-synaptic bursting (**Wittenberg and Wang, 2006**; **Inglebert et al., 2020**). In the cerebellum, synaptic plasticity rules with diverse temporal requirements on the time difference between parallel fiber and climbing fiber inputs have been found in Purkinje cells in different zones of this structure suvrathan16. While this heterogeneity has been found so far across structures or across different regions in the same structure, this heterogeneity could also be present within local networks, as current experimental methods for probing plasticity only have access to a single delay between pre and post-synaptic spikes in each recorded neuron, and would therefore miss this heterogeneity.

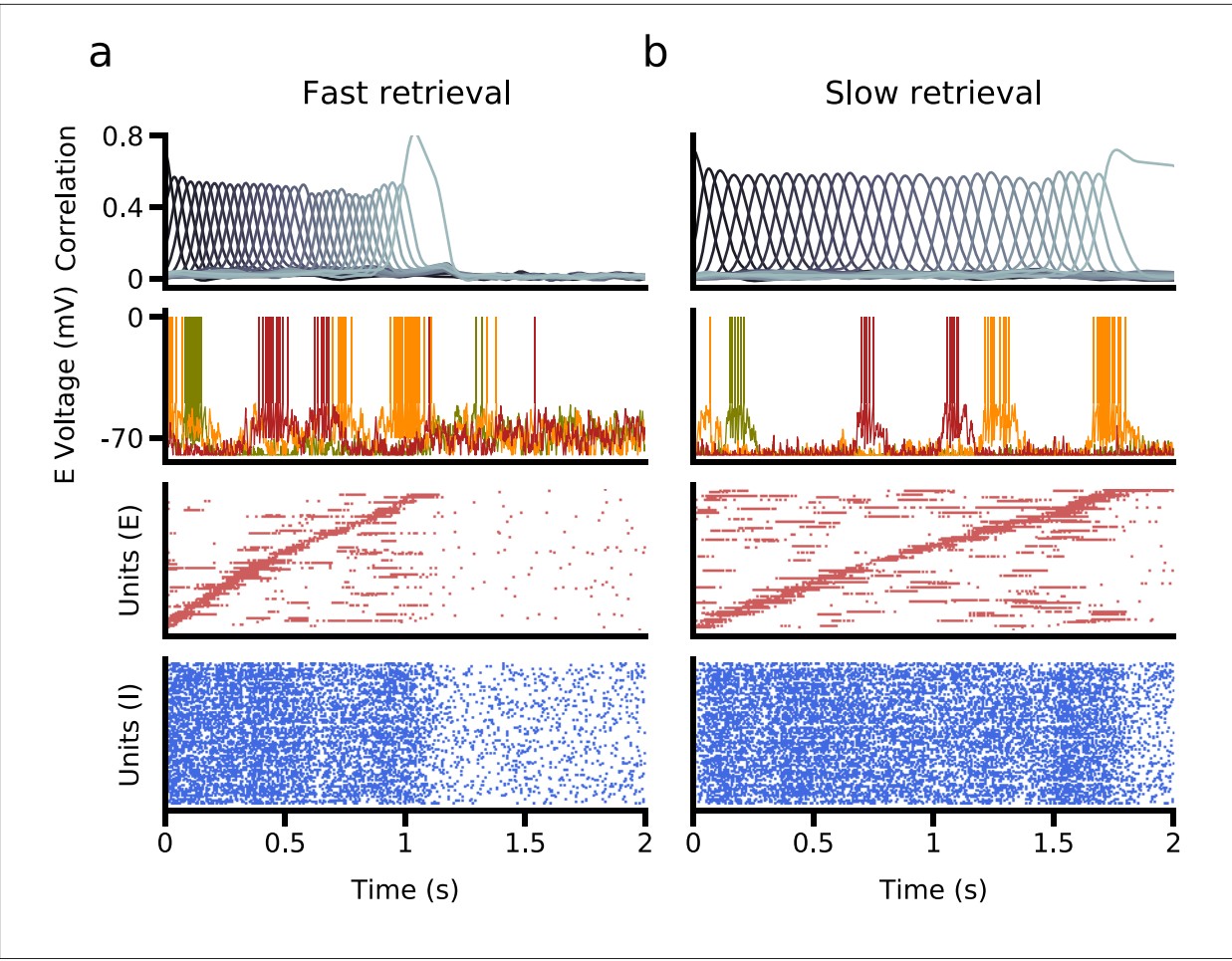

**Figure 7.** Retrieval in a network of excitatory and inhibitory spiking neurons, in which excitatory neurons are subdivided into asymmetric and symmetric populations. (**a**) Fast retrieval when inputs are biased to asymmetric neurons. *Top*: Correlation of network activity with each stored pattern. *Middle*: Voltage traces of three representative neurons. *Bottom*: The bottom two panels show raster plots of excitatory and inhibitory neurons, sorted by the latency of their peak firing rate. (**b**) Slow retrieval when inputs are biased to symmetric neurons. Note that only external inputs differ in (**a**) and (**b**). See Methods for parameters.

For simplicity, the degree of temporal asymmetry was chosen in our model to depend only on the identity of the postsynaptic neuron. This is consistent with the observation that a model of synaptic plasticity that depends only on the postsynaptic concentration of calcium can account for a range of experimentally observed STDP curves (*Graupner and Brunel, 2012*). This suggests that heterogeneities in temporal asymmetry could arise due to heterogeneities in biophysical parameters that control calcium dynamics in post-synaptic spines.

## Comparison with other mechanisms of speed control

The mechanism investigated here is distinct from previously described models of input-driven speed control. It does not require adaptation mechanisms or delays to slow down retrieval of subsequent patterns (*Sompolinsky and Kanter, 1986*; *Murray and Escola, 2017*). It also does not require presentation of multiple exemplars spanning the desired range of retrieval speeds in order to find the appropriate network structure (*Wang et al., 2018*). However, the mapping between external input strength and retrieval speed must be learned in order for the network to be able to perform retrieval at desired speeds. Unlike the model explored in *Wang et al., 2018*, however, once this mapping is learned, it can be used to control the speed of other stored sequences.

Another recent study (*Beiran et al., 2023*) has investigated how a recurrent network could flexibly control its temporal dynamics using a different approach. They trained a low-rank recurrent network using back-propagation through time to produce specific dynamics with flexible timing, and showed

that the resulting network can then be flexibly controlled by a one-dimensional input. It would be interesting to investigate whether the low-rank structure found in such a manner exhibits similarities with the synaptic connectivity structure in our model.

Future experimental work could analyze the evolution of neural activity across the training of interval timing tasks, and evaluate whether it is consistent with such a reinforcement-based rule.

## Experimental predictions

This mechanism presented here makes several predictions regarding the relationship between plasticity rules, external input, and the speed of network dynamics. One prediction is that retrieval speed could be modified by providing different external inputs to each population (asymmetric and symmetric). In vivo, these populations could be identified using the dependence of mean firing rates on speed of retrieval - neurons who increase their rates with slower/faster retrieval speeds would be predicted to be the symmetric/asymmetric neurons, respectively. Targeting one class of neurons or the other, using holographic techniques (see e.g. *Marshel et al., 2019*) would then be expected to increase or decrease the speed of retrieval. Another prediction is that these cells have distinct profiles of temporal asymmetry in their synaptic plasticity. The model presented here also predicts the existence of 'null' input directions, for which no change in retrieval speed is expected as external input is changed. When moving along these 'null' directions, single neurons would only be expected to change their temporal firing patterns, but without affecting the speed of retrieval.

## Transitions between persistent and sequential activity

Heterogeneity in the learning rule also provides a mechanism that enables input changes to drive transitions in activity states. An example of such a transition is frequently reported in primary motor cortex (M1) during delayed reaching tasks, where a preparatory period with persistent activity or ramping dynamics is followed by an execution period with transient, sequential dynamics (*Riehle and Requin, 1989*; *Li et al., 2016*). We demonstrated how an input change can gate such a transition in a simple network model composed of neurons with two distinct plasticity rules, the first temporally symmetric, and the second temporally asymmetric. At the start of the preparatory period, asymmetric neurons are inhibited, and a transient specific input elicits persistent activity in symmetric neurons. When inhibition is removed, asymmetric neurons become activated and drive a transition to sequential activity in both types of neurons.

Inhibitory gating has been previously hypothesized as a mechanism to control the initiation of execution period activity. Analysis of M1 activity suggests that local inhibitory interneurons do not engage in this gating, as putative inhibitory neurons do not appear to be preferentially active during the preparatory period compared to the execution period (*Kaufman et al., 2013*). However, this does not rule out the possibility that the necessary inhibition could arise from other external inputs to M1. It is also possible that inhibition may not be required at all. Effective silencing of the asymmetric neurons could occur by a reduction of excitatory input during the preparatory period. Recent work in mice suggests that thalamocortical interactions may be a potential candidate for driving the required transition. Recorded activity in motor thalamus during a reaching task shows that at movement onset, thalamus activity is negatively correlated with premotor activity, but positively correlated with activity in M1 (*Nashef et al., 2021*). In a separate directional licking task, thalamus projections were shown to be required for initiating cued movement, and mimicked presentation of the cue when optogenetically stimulated (*Inagaki et al., 2022*). An alternative model for transitions between preparatory and execution activity has recently been proposed (*Bachschmid-Romano et al., 2023*), in which external inputs trigger a switch between a preparatory state and a nearly orthogonal execution state. However, in the model of *Bachschmid-Romano et al., 2023*, the execution epoch is described by a single pattern, and any temporal dynamics within this epoch is inherited from external inputs, while in the present paper the temporal dynamics during the execution phase is generated by the recurrent connectivity structure.

## Limitations and future directions

We have focused here on a simple learning scenario in which a temporally asymmetric plasticity rule imprints a sequence of external input patterns into the recurrent synaptic connectivity. In real neuronal networks, one expects recurrent synaptic inputs to shape the response of a network to external

inputs, and therefore how such inputs sculpt recurrent connectivity. Studying such a learning process is outside the scope of this paper, but is an important topic for future work.

In this paper, we have focused on Hebbian specific synaptic plasticity rules to store a sequence of input patterns. Another fruitful approach to investigate learning and memory in neural circuits was introduced by *Gardner, 1988*. In Gardner's approach, the idea is to consider the space of all possible connectivity matrices that store a given set of memories as fixed point attractor states. It was later shown that the statistics of the connectivity matrix in attractor networks with sign-constrained synapses optimizing the storage capacity and/or robustness of learning is in striking agreement with cortical data - in particular, the resulting connectivity is sparse, with an overrepresentation of bidirectional motifs in pairs of neurons, compared to random directed Erdos-Renyi networks (*Brunel, 2016*). However, in networks storing sequences, no such overrepresentation exists (*Brunel, 2016*). It will be interesting to investigate the statistics of connectivity in networks with flexibility constraints, such that sequences can be retrieved at different speeds, or with a coexistence of fixed point attractor dynamics with sequential retrieval.

## Methods
### Neuronal transfer function
The neuronal transfer function is given by the sigmoidal function

$$\phi(h) = \frac{r_{\max}}{2}\left(1 + \mathrm{erf}\left(\frac{h - \theta}{\sqrt{2}\sigma}\right)\right) \tag{9}$$

where $\theta$ determines the input at which the neuron fires at half the maximal value $r_{max}$, and $\sigma$ is inversely proportional to the gain. This function was chosen for continuity with previous work (*Gillett et al., 2020*). We expect that using qualitatively similar functions should not alter the results of this paper.

### Noisy inputs
We introduce noisy inputs to each neuron in *Figures 1e and 2c* through independent realizations of an Ornstein-Uhlenbeck process with a mean equal to either $I_a^{\mathrm{ext}}$ or $I_s^{\mathrm{ext}}$, respectively, with standard deviation of 0.3, and a correlation time constant of 4 ms. This noise leads to fluctuations of firing rate that are comparable to rate fluctuations induced by sequence retrieval (*Figure 2c*), while leaving sequence retrieval intact (*Figure 1e*).

### Measuring pattern correlations
To compute the Pearson pattern correlation $m_\mu(t)$, we compute the overlap of each of the stored patterns $\xi_\mu$ with the instantaneous firing rates for the entire population and divide by the standard deviation of firing rate activity: $m_\mu(t) = \frac{1}{N}\sum_{i=1}^{N} r_i \xi_i^\mu / \sigma_r(t)$. In *Figures 3 and 4*, we compute the correlations separately for each subpopulation.

### Measuring retrieval speed
To measure retrieval speed $v$ in *Figures 2, 3 and 5*, we recorded the times at which each pattern correlation attained its peak value, and computed the average time difference between the peaks of successive patterns in a sequence. We then divided the time constant of the rate dynamics by this averaged value in order to convert speed into units of $\tau^{-1}$:

$$v = \frac{\tau}{\frac{1}{P-1}\sum_{l=2}^{P} argmax_t\big(m_l(t)\big) - argmax_t\big(m_{l-1}(t)\big)} \tag{10}$$

To account for simulations with dynamics that did not have well-defined correlation peaks (typically observed at extreme storage loads or with persistent activity), we excluded peak time differences that exceeded two standard deviations of the average difference value. If no peak time difference passed this criteria, the sequence was considered not retrieved (black regions in *Figures 2, 3 and 5*).

## Mean-field theory of single-population network with variable degree of temporal asymmetry

In this section we derive a mean-field theory for the single population network with homogeneous synaptic plasticity. This is a generalization of the theory derived for a purely temporally asymmetric network *Gillett et al., 2020*. We define order parameters $q_\mu(t) = \mathbb{E}\left(r(t)\xi^\mu\right)$ and $M = \mathbb{E}\left((r)^2(t)\right)$, describing the average overlap of network activity with pattern $\xi^\mu$ and the average squared firing rate, respectively.

Using *Equations 1 and 2*, we derive equations describing the temporal evolution of the overlaps (for $l \in \{2, ..., P\}$),

$$\tau \frac{dq_l}{dt} = -q_l + \int D\xi^l Dx \xi^l \phi\left((1-z)\xi^l q_{l-1} + z\xi^l q_l + R_l x + I^{\text{ext}}\right) \tag{11}$$

where $R_l$ is a 'noise' term due to patterns $\mu \neq l$ in the sequence (see *Gillett et al., 2020* for details) By making the following change of variables:

$$v = \frac{q_l^{\tilde{z}}\xi_l + xR_l}{\sqrt{(q_l^{\tilde{z}})^2 + R_l^2}} \tag{12}$$

$$u = \frac{\xi_l R_l - q_l^{\tilde{z}}x}{\sqrt{(q_l^{\tilde{z}})^2 + R_l^2}} \tag{13}$$

in which we have defined $q_l^{\tilde{z}} = (1-z)q_{l-1} + zq_l$, we obtain

$$\tau \frac{dq_l}{dt} = -q_l + q_l^{\tilde{z}}G\left((q_l^{\tilde{z}})^2 + R_l^2\right), \tag{14}$$

where

$$G(x) = \frac{\int Dv\phi\left(v\sqrt{x} + I_{ext}\right)}{\sqrt{x}}. \tag{15}$$

Assuming that $G(x) \approx 1$, which is the case during successful retrieval (see also *Gillett et al., 2020*), then we can simplify to:

$$\frac{\tau}{1-z} \frac{dq_l}{dt} = -q_l + q_{l-1} \tag{16}$$

This equation makes it clear that retrieval speed depends linearly on $z$, that is on the balance between the symmetric and asymmetric components of synaptic plasticity.

## Mean-field theory of heterogeneous network and conditions for retrieval

Mean-field theory can be used to further analyze retrieval speed dynamics, along the lines of *Gillett et al., 2020*. We define order parameters $q_\mu^X(t) = \mathbb{E}\left(r^X(t)\xi^{X,\mu}\right)$ and $M_X = \mathbb{E}\left((r^X)^2(t)\right)$, describing the average overlap of network activity in subpopulation $X$ with pattern $\xi^{X,\mu}$ and the average squared firing rate in subpopulation $X$, respectively. The equations for the overlaps are given by:

$$\tau \frac{dq_l^a}{dt} = -q_l^a + \left(q_{l-1}^a + q_{l-1}^s\right) \cdot G_a\left((q_{l-1}^a + q_{l-1}^s)^2 + (R_l^a)^2\right) \tag{17}$$

$$\tau \frac{dq_l^s}{dt} = -q_l^s + \left(q_l^s + q_l^a\right) \cdot G_s\left((q_l^a + q_l^s)^2 + (R_l^s)^2\right) \tag{18}$$

where G is given, for arbitrary transfer functions $\phi$ by:

$$G_X(x) = \frac{\int Dv\phi\left(v\sqrt{x} + I_X^{ext}\right)}{\sqrt{x}}. \tag{19}$$

For the transfer function used in this paper, *Equation 9*, the expression simplifies,

$$G_X(x) \equiv \frac{1}{\sqrt{2\pi(\sigma^2 + x)}} \exp\left(-\frac{(\theta + I_X^{ext})^2}{2(\sigma^2 + x)}\right). \tag{20}$$

As in the previous section, $R_l^a$ and $R_l^s$ are 'noise' terms due to patterns $\mu \neq l$ in the sequence, which also depends on the average squared firing rates $M_a$ and $M_s$. Using **Equations 17 and 18**, we can derive the dynamics of the combined population overlap $q_l(t) = q_l^a(t) + q_l^s(t)$:

$$\tau \frac{dq_l}{dt} = -q_l + q_{l-1}G_a\left((q_{l-1})^2 + (R_l^a)^2\right) + q_l G_s\left((q_l)^2 + (R_l^s)^2\right) \tag{21}$$

To compute the boundary for successful retrieval given by the white line in **Figure 2**, we analyze this equation when the gains are constant: $G_x(t) = G_x$. Plugging in and rearranging, we find:

$$\frac{\tau}{1 - G_s}\frac{dq_l}{dt} = -q_l + \frac{G_a}{1 - G_s}q_{l-1} \tag{22}$$

This equation shows that the sequence can only be retrieved if $G_a/(1 - G_s) > 1$, otherwise the peak of the overlaps decay to zero with increasing $l$. Thus retrieval of an asymptotically long sequence is successful if the gain converges to a value greater or equal to one during retrieval. This condition can only be satisfied if

$$\max_x \left[G_a(x; \theta, \sigma, I_{ext}^a) + G_s(x; \theta, \sigma, I_{ext}^s)\right] \geq 1 \tag{23}$$

To test for successful sequence retrieval in **Figure 2**, we computed the maximal correlation value of the final pattern $m_P(t)$, and compared this value to a threshold $\theta_P = 0.05$. If the value fell below this threshold, then retrieval was considered unsuccessful, and was denoted by a black square. This threshold criterion was also used in **Figures 3 and 5**.

## Reward-driven learning

A simple perturbation-based reinforcement learning rule is used to demonstrate that external inputs can be generated that produce network dynamics at a desired target speed over the course of multiple trial repetitions. We simulate a series of trials with stochastically varying external inputs. At each trial $n$, the external inputs used in the previous trial are perturbed randomly,

$$I_{\text{pert}}^{ext,a} = I_{n-1}^{ext,a} + \lambda \Delta x_n^a \tag{24}$$

$$I_{\text{pert}}^{ext,s} = I_{n-1}^{ext,s} + \lambda \Delta x_n^s \tag{25}$$

where $\lambda$ is the strength of the perturbation, and $\Delta x_n^p$ are uniformly distributed random variables over the interval $[-1, 1]$, drawn independently for each population $p \in \{a, s\}$ at each trial $n$. If these external inputs lead to an improvement in speed compared to previous trials, then

$$I_n^{ext,a} = I_{\text{pert}}^{ext,a} \tag{26}$$

$$I_n^{ext,s} = I_{\text{pert}}^{ext,s} \tag{27}$$

else,

$$I_n^{ext,a} = I_{n-1}^{ext,a} \tag{28}$$

$$I_n^{ext,s} = I_{n-1}^{ext,s} \tag{29}$$

In **Figure 6**, the correlation threshold $\theta_m = 0.05$, the target speed $v_{target} = \{0.3, 0.8\}$, and $\lambda = 0.1$. On the first trial ($n = 0$), the external inputs are taken to be $I_0^{ext,a} = -0.2$ and $I_0^{ext,s} = -0.2$ (open circle in **Figure 6**).

## Network of excitatory and inhibitory spiking neurons

We simulated a network of excitatory and inhibitory leaky integrate-and-fire (LIF) neurons similar to the one described described in the Appendix of **Gillett et al., 2020** (sections 3 and 4) with a few differences described below.

In this network, the dynamics of the membrane potential of neuron $i$ ($i = 1, \ldots, N_\alpha$) in population $\alpha$ ($\alpha = E, I$) are governed by the following equations:

$$\tau_m^\alpha \frac{dV_i^\alpha}{dt} = \Theta\left(V_i^\alpha - V_{\text{floor}}^\alpha\right) \left(-V_i^\alpha + V_{\text{rest}}^\alpha + \sum_\beta \sum_{j \neq i}^{K_{\alpha\beta}} S_{ij}^{\alpha\beta} + I_i^\alpha(t) + \sigma^\alpha \sqrt{\tau_m^\alpha} W^\alpha(t)\right) \tag{30}$$

$$\tau_s^\alpha \frac{dS_{ij}^{\alpha\beta}}{dt} = -S_{ij}^{\alpha\beta} + J_{ij}^{\alpha\beta} \tau_s^\alpha \sum_{t_j^k} \delta\left(t - t_j^k - D\right) \tag{31}$$

where $\alpha, \beta \in \{E, I\}$, $D$ controls the synaptic delay, $I_\alpha(t)$ controls the time-dependent external input drive, $\tau_{rp}$ controls the refractory period, $\Theta(x)$ is the Heaviside function, and $W_\alpha(t)$ is a white noise input with zero mean and unit variance density.

Excitatory neurons are divided into two (asymmetric and symmetric) populations of equal size ($N_E = N_a + N_s$), with connectivity matrices given by the following, where $\omega$ is the rectified synaptic transfer function defined in the procedure and $X \in \{a, s\}$:

$$J_{ij}^{aX} = \frac{c_{ij}^{aX}}{\sqrt{N_a c}} \omega\left(\frac{A_{EE}}{\sqrt{N_a c}} \sum_{\mu=1}^{P-1} f(\xi_i^{a,\mu+1}) g(\xi_j^{X,\mu})\right) \tag{32}$$

$$J_{ij}^{sX} = \frac{c_{ij}^{sX}}{\sqrt{N_s c}} \omega\left(\frac{A_{EE}}{\sqrt{N_s c}} \sum_{\mu=1}^{P-1} f(\xi_i^{s,\mu}) g(\xi_j^{X,\mu})\right). \tag{33}$$

The excitatory populations receive external input that depends on their identity, and on the retrieval configuration. For slow retrieval, we set the input $I_E^i(t)$ equal to $I_a^{ext} = -1.5\text{mV}$ and $I_s^{ext} = 0.5\text{mV}$ for asymmetric and symmetric neurons, respectively. For fast retrieval, we use $I_a^{ext} = 0.75\text{mV}$ and $I_s^{ext} = -0.75\text{mV}$. In inhibitory neurons, we use $I_i^i(t) = 0$.

The learning strength ($A_{EE}$) is set to .25, which result in changes to the following parameters: $J^{IE}/K_{IE} = 0.134$, $\lambda_V^E = 5.123$, $\lambda_V^I = 3.012$. All other parameter values are identical to those documented in Table 7c of the referenced Appendix (*Gillett et al., 2020*).

---

# Additional information

### Funding

| Funder | Grant reference number | Author |
|---|---|---|
| National Institutes of Health | R01 EB022891 | Maxwell Gillett Nicolas Brunel |
| Office of Naval Research | N00014-16-1-2327 | Maxwell Gillett Nicolas Brunel |

The funders had no role in study design, data collection and interpretation, or the decision to submit the work for publication.

### Author contributions

Maxwell Gillett, Conceptualization, Software, Formal analysis, Investigation, Writing - original draft, Writing - review and editing; Nicolas Brunel, Conceptualization, Formal analysis, Writing - review and editing

### Author ORCIDs

Maxwell Gillett ⓘ https://orcid.org/0000-0002-1937-477X
Nicolas Brunel ⓘ https://orcid.org/0000-0002-2272-3248

Reviewer #1 (Public Review): https://doi.org/10.7554/eLife.88805.3.sa1
Author response https://doi.org/10.7554/eLife.88805.3.sa2

---

# Additional files

## Supplementary files
• MDAR checklist

## Data availability
The current manuscript is a computational study, so no data have been generated for this manuscript. Code to run the simulations and build the figures have been uploaded to Github at https://github.com/maxgillett/dynamic_speed_control, (copy archived at *Gillett, 2024*).

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
