## [Editor Report · eLife assessment]

The authors provide a **valuable** analysis of what neural circuit mechanisms enable varying the speed of retrieval of sequences, which is needed in situations such as reproducing motor patterns. Their use of heterogeneous plasticity rules to allow external currents to control speed of sequence recall is a novel alternative to other mechanisms proposed in the literature. They perform a **convincing** characterization of relevant properties of recall via simulations and theory, though a better mapping to biologically plausible mechanisms is left for future work.

---

## [Referee Report · Reviewer #1 (Public Review)]

While there are many models for sequence retrieval, it has been difficult to find models that vary the speed of sequence retrieval dynamically via simple external inputs. While recent works have proposed some mechanisms, the authors here propose a different one based on heterogeneous plasticity rules. Temporally symmetric plasticity kernels (that do not distinguish between the order of pre and post spikes, but only their time difference) are expected to give rise to attractor states, asymmetric ones to sequence transitions. The authors incorporate a rate-based, discrete-time analog of these spike-based plasticity rules to learn the connections between neurons (leading to connections similar to Hopfield networks for attractors and sequences). They use either a parametric combination of symmetric and asymmetric learning rules for connections into each neuron, or separate subpopulations having only symmetric or asymmetric learning rules on incoming connections. They find that the latter is conducive to enabling external inputs to control the speed of sequence retrieval.

Comments on revised version:

The authors have addressed most of the points of the reviewers.

A major substantive point raised by both reviewers was on the biological plausibility of the learning.

The authors have added a section in the Discussion. This remains an open question, however the discussion suffices for the current paper.

---

## [Author Response]

Author response:

The following is the authors’ response to the original reviews.

**Reviewer #1 (Public Review):**
While there are many models for sequence retrieval, it has been difficult to find models that vary the speed of sequence retrieval dynamically via simple external inputs. While recent works [1,2] have proposed some mechanisms, the authors here propose a different one based on heterogeneous plasticity rules. Temporally symmetric plasticity kernels (that do not distinguish between the order of pre and post spikes, but only their time difference) are expected to give rise to attractor states, asymmetric ones to sequence transitions. The authors incorporate a rate-based, discrete-time analog of these spike-based plasticity rules to learn the connections between neurons (leading to connections similar to Hopfield networks for attractors and sequences). They use either a parametric combination of symmetric and asymmetric learning rules for connections into each neuron, or separate subpopulations having only symmetric or asymmetric learning rules on incoming connections. They find that the latter is conducive to enabling external inputs to control the speed of sequence retrieval.Strengths:The authors have expertly characterised the system dynamics using both simulations and theory. How the speed and quality of retrieval varies across phases space has been well-studied. The authors are also able to vary the external inputs to reproduce a preparatory followed by an execution phase of sequence retrieval as seen experimentally in motor control. They also propose a simple reinforcement learning scheme for learning to map the two external inputs to the desired retrieval speed.Weaknesses:(1) The authors translate spike-based synaptic plasticity rules to a way to learn/set connections for rate units operating in discrete time, similar to their earlier work in [5]. The bio-plausibility issues of learning in [5] carry over here, for e.g. the authors ignore any input due to the recurrent connectivity during learning and effectively fix the pre and post rates to the desired ones. While the learning itself is not fully bio-plausible, it does lend itself to writing the final connectivity matrix in a manner that is easier to analyze theoretically.

We agree with the reviewer that learning is not `fully bio-plausible’. However, we believe that extending the results to a model in which synaptic plasticity depends on recurrent inputs is beyond the scope of this work. We have added a mention of this issue in the Discussion in the revised manuscript.

(2) While the authors learn to map the set of two external input strengths to speed of retrieval, they still hand-wire one external input to the subpopulation of neurons with temporally symmetric plasticity and the other external input to the other subpopulation with temporally asymmetric plasticity. The authors suggest that these subpopulations might arise due to differences in the parameters of Ca dynamics as in their earlier work [29]. How these two external inputs would connect to neurons differentially based on the plasticity kernel / Ca dynamics parameters of the recurrent connections is still an open question which the authors have not touched upon.

The issue of how external inputs could self-organize to drive the network to retrieve sequences at appropriate speeds is addressed in the Results section, paragraph `Reward-driven learning’. These inputs are not `hand-wired’ - they are initially random and then acquire the necessary strengths to allow the network to retrieve the sequences at different speeds thanks to a simple reinforcement learning scheme. We have rewritten this section to clarify this issue.

(3) The authors require that temporally symmetric and asymmetric learning rules be present in the recurrent connections between subpopulations of neurons in the same brain region, i.e. some neurons in the same brain region should have temporally symmetric kernels, while others should have temporally asymmetric ones. The evidence for this seems thin. Though, in the discussion, the authors clarify 'While this heterogeneity has been found so far across structures or across different regions in the same structure, this heterogeneity could also be present within local networks, as current experimental methods for probing plasticity only have access to a single delay between pre and post-synaptic spikes in each recorded neuron, and would therefore miss this heterogeneity'.

We agree with the reviewer that this is currently an open question. We describe this issue in more detail in the Discussion of the revised manuscript.

(4) An aspect which the authors have not connected to is one of the author's earlier work:Brunel, N. (2016). Is cortical connectivity optimized for storing information? Nature Neuroscience, 19(5), 749-755 https://doi.org/10.1038/nn.4286 which suggests that the experimentally observed over-representation of symmetric synapses suggests that cortical networks are optimized for attractors rather githubhan sequences.

We thank the reviewer for this suggestion. We have added a paragraph in the discussion that discusses work on statistics of synaptic connectivity in optimal networks. We expect that in networks that contain two subpopulations of neurons, the degree of symmetry should be intermediate between a network storing fixed point attractors exclusively, and a network storing sequences exclusively.

Despite the above weaknesses, the work is a solid advance in proposing an alternate model for modulating speed of sequence retrieval and extends the use of well-established theoretical tools. This work is expected to spawn further works like extending to a spiking neural network with Dale's law, more realistic learning taking into account recurrent connections during learning, and experimental follow-ups. Thus, I expect this to be an important contribution to the field.

We thank the reviewer for the insightful comments.

**Reviewer #2 (Public Review):**
Sequences of neural activity underlie most of our behavior. And as experience suggests we are (in most cases) able to flexibly change the speed for our learned behavior which essentially means that brains are able to change the speed at which the sequence is retrieved from the memory. The authors here propose a mechanism by which networks in the brain can learn a sequence of spike patterns and retrieve them at variable speed. At a conceptual level I think the authors have a very nice idea: use of symmetric and asymmetric learning rules to learn the sequences and then use different inputs to neurons with symmetric or asymmetric plasticity to control the retrieval speed. The authors have demonstrated the feasibility of the idea in a rather idealized network model. I think it is important that the idea is demonstrated in more biologically plausible settings (e.g. spiking neurons, a network with exc. and inh. neurons with ongoing activity).SummaryIn this manuscript authors have addressed the problem of learning and retrieval sequential activity in neuronal networks. In particular, they have focussed on the problem of how sequence retrieval speed can be controlled?They have considered a model with excitatory rate-based neurons. Authors show that when sequences are learned with both temporally symmetric and asymmetric Hebbian plasticity, by modulating the external inputs to the network the sequence retrieval speed can be modulated. With the two types of Hebbian plasticity in the network, sequence learning essentially means that the network has both feedforward and recurrent connections related to the sequence. By giving different amounts of input to the feed-forward and recurrent components of the sequence, authors are able to adjust the speed.Strengths- Authors solve the problem of sequence retrieval speed control by learning the sequence in both feedforward and recurrent connectivity within a network. It is a very interesting idea for two main reasons: 1. It does not rely on delays or short-term dynamics in neurons/synapses 2. It does not require that the animal is presented with the same sequences multiple times at different speeds. Different inputs to the feedforward and recurrent populations are sufficient to alter the speed. However, the work leaves several issues unaddressed as explained below.Weaknesses- The main weakness of the paper is that it is mostly driven by a motivation to find a computational solution to the problem of sequence retrieval speed. In most cases they have not provided any arguments about the biological plausibility of the solution they have proposed e.g.:- Is there any experimental evidence that some neurons in the network have symmetric Hebbian plasticity and some temporally asymmetric? In the references authors have cited some references to support this. But usually the switch between temporally symmetric and asymmetric rules is dependent on spike patterns used for pairing (e.g. bursts vs single spikes). In the context of this manuscript, it would mean that in the same pattern, some neurons burst and some don't and this is the same for all the patterns in the sequence. As far as I see here authors have assumed a binary pattern of activity which is the same for all neurons that participate in the pattern.

There is currently only weak evidence for heterogeneity of synaptic plasticity rules within a single network, though there is plenty of evidence for such a heterogeneity across networks or across locations within a particular structure (see references in our Discussion). The reviewer suggests another interesting possibility, that the temporal asymmetry could depend on the firing pattern on the post-synaptic neuron. An example of such a behavior can be found in a paper by Wittenberg and Wang in 2006, where they show that pairing single spikes of pre and post-synaptic neurons lead to LTD at all time differences in a symmetric fashion, while pairing a pre-synaptic spike with a burst of post-synaptic spikes lead to temporally asymmetric plasticity, with a LTP window at short positive time differences. We now mention this possibility in the Discussion, but we believe exploring fully this scenario is beyond the scope of the paper.

- How would external inputs know that they are impinging on a symmetric or asymmetric neuron? Authors have proposed a mechanism to learn these inputs. But that makes the sequence learning problem a two stage problem -- first an animal has to learn the sequence and then it has to learn to modulate the speed of retrieval. It should be possible to find experimental evidence to support this?

Our model does not assume that the two processes necessarily occur one after the other. Importantly, once the correct external inputs that can modulate sequence retrieval are learned, sequence retrieval modulation will automatically generalize to arbitrary new sequences that are learned by the network.

- Authors have only considered homogeneous DC input for sequence retrieval. This kind of input is highly unnatural. It would be more plausible if the authors considered fluctuating input which is different from each neuron.

We have modified Figure 1e and Figure 2c to show the effects of fluctuating inputs on pattern correlations and single unit activity. We find that these inputs do not qualitatively affect our results.

- All the work is demonstrated using a firing rate based model of only excitatory neurons. I think it is important that some of the key results are demonstrated in a network of both excitatory and inhibitory spiking neurons. As the authors very well know it is not always trivial to extend rate-based models to spiking neurons.I think at a conceptual level authors have a very nice idea but it needs to be demonstrated in a more biologically plausible setting (and by that I do not mean biophysical neurons etc.).

We have included a new section in the discussion with an associated figure (Figure 7) demonstrating that flexible speed control can be achieved in an excitatory-inhibitory (E-I) spiking network containing two excitatory populations with distinct plasticity mechanisms.

**Reviewer #1 (Recommendations For The Authors):**
In the introduction, the authors state: 'symmetric kernels, in which coincident activity leads to strengthening regardless of the order of pre and post-synaptic spikes, have also been observed in multiple contexts with high frequency plasticity induction protocols in cortex [21]'. To my understanding, [21]'s final model 3, ignores LTD if the post-spike also participates in LTP, and only considers nearest-neighbour interactions. Thus, the kernel would not be symmetric. Can the authors clarify what they mean and how their conclusion follows, as [21] does not show any kernels either.

In this statement, we were not referring to the model in [21], but rather the experimentally observed plasticity kernels at different frequencies. In particular, we were referring to the symmetric kernel that appears in the bottom panel of Figure 7c in that paper.

The authors should also address the weaknesses mentioned above. They don't need to solve the issues but expand (and maybe indicate resolutions) on these issues in the Discussion.For ease of reproducibility, the authors should make their code available as well.

We intend to publish the code required to reproduce all figures on Github.

**Reviewer #2 (Recommendations For The Authors):**
- Show the ground state of the network before and after learning.

We have decided not to include such a figure, as we have not analyzed the learning process, but instead a network with a fixed connectivity matrix which is assumed to be the end result of a learning process.

- Authors have only considered a network of excitatory neurons. This does not make sense. I think they should demonstrate a network of both exc. and inch. neurons (spiking neurons) exhibiting ongoing activity.

See our comment to Reviewer #2 in the previous section.

- Show how the sequence dynamics unfolds when we assume a non-zero ongoing activity.

We are not sure what the reviewer means by `non-zero ongoing activity. We show now the dynamics of the network in the presence of noisy inputs, which can represent ongoing activity from other structures (see Fig 1e and 2c).

- From the correlation (==quality) alone it is difficult to judge how well the sequence has been recovered. Authors should consider showing some examples so that the reader can get a visual estimate of what 0.6 quality may mean. High speed is not really associated with high quality (Fig 2b). So it is important to show how the sequence retrieval quality is for non-linear and heterogeneous learning rules.

We believe that some insight into the relationship between speed and quality for the case of non-linear and heterogeneous learning rules is addressed by the correlation plots for chosen input configurations (see Fig. 3a and 5b). We leave a full characterization for future work.

- Authors should show how the retrieval and quality of sequences change when they are recovered with positive input, or positive input to one population and negative to another. In the current version sequence retrieval is shown only with negative inputs. This is a somewhat non-biological setting. The inhibitory gating argument (L367-389) is really weak.

We would like to clarify that with the parameters chosen in this paper, the transfer function has half its maximal rate at zero input. This is due to the fact we chose the threshold to be zero, using the fact that any threshold can be absorbed in the external inputs. Thus, negative inputs really mean sub-threshold inputs, and they are consistent with sub-threshold external excitatory inputs. We have clarified this issue in the revised manuscript.

- Authors should demonstrate how the sequence retrieval dynamics is altered when they assume a fluctuating input current for sequence retrieval instead of a homogeneous DC input.

See our comment to Reviewer #2 in the previous section.

- Authors should show what are the differences in synaptic weight distribution for the two types of learning (bi-linear and non-linear). I am curious to know if the difference in the speed in the two cases is related to the weight distribution. In general I think it is a good idea to show the synaptic weight distribution before and after learning.

As mentioned above, we do not study any learning process, but rather a network with a fixed connectivity matrix, assumed to represent the end result of learning. In this network, the distribution of synaptic weights converges to a Gaussian in the large p and cN limits, independently of the functions f and g, because of the central limit theorem, if there are no sign constraints on weights. In the presence of sign constraints, the distribution is a truncated Gaussian.

- I suggest the use of a monochromatic color scale for figure 2b and 3b.Figure 3: The sentence describing panel 2 seems incomplete.Also explain why there is non-monotonic relationship between I_s and speed for some values ofI_a in 3b

There is a non-monotonic relationship for retrieval quality, not speed. We have clarified this in the manuscript text, but don’t currently have an explanation for why this phenomenon occurs for these specific values of I_a.